# ESD Ideas: Propagation of high-frequency forcing to ice age dynamics

*Mikhail Y. Verbitsky[1,a,\*], Michel Crucifix[2], and Dmitry M. Volobuev[3]*

[1]Gen5 Group, LLC, Newton, MA, USA
[2]UCLouvain, Earth and Life Institute, Louvain-la-Neuve, Belgium
[3]The Central Astronomical Observatory of the Russian Academy of Sciences at Pulkovo, Saint Petersburg, Russia
[a]formerly at: Yale University, Department of Geology and Geophysics, New Haven, CT, USA
[*]retired
Correspondence: Mikhail Verbitsky (verbitskys@gmail.com)

**Abstract.** Palaeoclimate records display a continuous background of variability connecting centennial to 100-ky periods. Hence, the dynamics at the centennial, millennial, and astronomical time scales should not be treated apart. Here, we show that the non-linear character of ice sheet dynamics, which was derived naturally from the ice-flow conservation laws, provides the scaling constraints to explain the structure of the observed spectrum of variability.

**Introduction.** Most theories of Quaternary climates consider that glacial-interglacial cycles emerge from components of the climate system interacting with each other, and responding to the forcing generated by the variations of summer insolation caused by climatic precession, the changes in obliquity, and in eccentricity. A common approach is to represent these interactions and response by ordinary differential equations. In a low-order dynamical system, the state vector only includes a handful of variables, which vary on roughly the same time scales as the forcing. Barry Saltzman has long promoted this approach, and his models state variables represented the volume of continental ice sheets, deep ocean temperature, carbon dioxide concentration, and in some models the lithospheric depression (e.g., Saltzman and Verbitsky, 1993). Similar models featuring other mechanisms were published more recently (e.g., Omta et al., 2016). The purpose of these models is to explain the temporal structure of ice age cycles, but the spectrum of variability at centennial and millennial time scales is generally ignored. This approach is commonly justified by a hypothesis of separation of time scales, as formulated by Saltzman (1990). However, this hypothesis is questionable. Indeed, the observational records display a continuous background of variability connecting centennial to 100-ky periods (Huybers and Curry, 2006). For this reason, the dynamics at the centennial, millennial, and astronomical time scales should not be considered separately. Here, we address this concern and show that the ice dynamics is an effective vehicle for propagating high-frequency forcing upscale.

**Methods.** To make this case, we use the dynamical model previously presented in Verbitsky et al. (2018). This non-linear dynamical system was derived from scaled conservation equations of ice flow, combined with an equation describing the evolution of a variable synthesizing the state of rest of the climate, called "climate temperature". The three variables are thus the area of glaciation, ice sheet basal temperature, and climate temperature. Without astronomical forcing, the system evolves to an equilibrium. When the astronomical forcing is present, the system exhibits different modes of non-linearity leading to different periods of ice-age rhythmicity. Specifically, when the ratio of positive climate feedback to negative glaciation feedback (quantified by the *V*-number) is about 0.75, the system displays glacial-interglacial cycles of a period of roughly 100-ky. In effect, the response doubles the obliquity period. For this mechanism to operate, ice needs to survive through a first maximum of insolation, and then grow to a level at which it is vulnerable to an - even modest - increase in insolation. In the Verbitsky et al. (2018) model with reference parameters, the threshold corresponds to a glaciation area $S$ of roughly $20 \times 10^6$ km[2].

In the reference experiment presented in Verbitsky et al.(2018) the system is driven, following standard practice, by mid-June insolation at 65°N (Berger and Loutre, 1991). The output of three additional experiments is shown here. In the first experiment, the mid-June insolation is replaced with a sinusoid of 5-ky period and variable amplitude (first, about the same amplitude as of insolation forcing, and then increased tenfold). In the second experiment, this tenfold increased 5-ky-period sinusoid is combined with mid-June insolation at 65°N. In the third experiment, the forcing is represented by several sinusoids of smaller amplitudes (~2.5 of the insolation forcing amplitude) and periods spread between 3 ky and 9 ky. The results demonstrate the following:

A. When our system is forced by a pure 5-ky sinusoid of small amplitude, the system remains in the vicinity of its equilibrium point, with glaciation area of $15 \times 10^6$ km$^2$ and climate temperature of 2°C (cf. Fig 1(A)). When the amplitude of the sinusoid is increased tenfold, the effects of the negative phases of the forcing no longer symmetric to those of the positive phases, because of the system's non-linearity. As a consequence, the system moves to a different phase-plane domain, around $6 \times 10^6$ km$^2$ of glaciation area and climate temperature of 4.6°C (Fig 1(A)).

B. This shift of the time-mean glaciation area and temperature has a dramatic effect on ice-age dynamics. When insolation forcing is combined with strong millennial forcing, the latter moves the system into the domain where obliquity-period doubling no longer occurs, because ice no longer grows to the level needed to enable the strong positive deglaciation feedback. Consequently, the 100-ky variability almost vanishes - Fig. 1(B). We term "hijacking", this suppression of ice age variability by millennial variability. This result by itself invalidates the classical time-scale separation hypothesis: we see here that increased *millennial* variability causes the *ice age* cycles to fade.

C. Millennial forcings can be aggregated: Several sinusoids of smaller amplitudes and of different millennial periods create the same "hijacking" effect as a single 5-ky high-amplitude sinusoid, moving the system into the phase-plane domain of higher temperatures and lower ice volume (Fig.1(C)).

D. Acting alone, low-amplitude millennial sinusoids preserve their original frequencies. However, several components may generate low-frequency beatings, which are then demodulated by the system. Through this mechanism, millennial forcing may induce responses at periods close to the orbital periods, e.g., periods of precession and obliquity (Fig. 1(D)). For example, 41-ky mode in Fig. 1D, which one might be tempted to attribute to obliquity, is in fact the demodulation of the envelope generated by the interplay of the 6-ky and 7-ky forcing sinusoids: $1/41 \approx 1/6 - 1/7$.

It is possible to anticipate the disruptive effect of forcing at other periods. Indeed, let us measure this disruption potential as the distance $\Delta S$ (km$^2$) on the phase plane, between the system's equilibrium point with zero forcing, and the time-mean ice-sheet area expected given a periodic forcing of amplitude $\varepsilon$ and period $T$ (Fig. 1 (A, C)). In Verbitsky et al. (2018), we have shown that the dynamical properties of the system are largely determined by the $V$-number. We therefore may expect $\Delta S = \varphi$ $(V, \varepsilon, T)$. Since $V$ is dimensionless, and since the dimensions of $\varepsilon$ (km/ky) and $T$ (ky) are independent (in our model, the forcing is introduced as a component of ice sheet mass balance and therefore $\varepsilon$ has the same dimension as ice ablation rate, km/ky), the $\pi$-theorem (Buckingham, 1914) tells us that $\Delta S/(\varepsilon^2 T^2) = \Phi(V)$. We determined experimentally that $\Delta S = 0$ if $V = 0$, and that $\Phi(V)$ can be approximated as a linear function. The scaling argument finally brings us to:

$$\Delta S \approx -\mu V \varepsilon^2 T^2 = -\mu V \varepsilon^2 f^{-2} \qquad (1)$$

where $f = 1/T$ is the frequency, and $\mu$ is a constant that has to be determined experimentally. We thus see that $\Delta S \sim f^{-2}$. The "-2" frequency slope of $\Delta S$ has been confirmed in additional numerical experiments (not shown here) for forcing periods between 2 ky and 20 ky. The 5-ky-period sinusoids and multiple sinusoids of periods spread between 3-ky and 9-ky are arbitrary choices used to illustrate the "hijacking" and beating effects. The phenomena can be replicated with other modes of millennial activity such as, e.g., 6.5-ky, 2.5-ky, 0.9-ky, and 0.5-ky periods identified by Dima and Lohmann (2009). For example, we confirmed that, as it is implied by equation (1), the "hijacking" effect of a 6.5-ky sinusoid is the same as of a 5-ky sinusoid if the ratio of the corresponding amplitudes is 0.77.

Similar scaling arguments can be applied to the amplitude of $S$-variable, i.e. amplitude of the glacial area over a glacial cycle. This amplitude $\bar{S}$ has the same dimension as $\Delta S$, i.e., $\bar{S} = \psi$ $(V, \varepsilon, T)$ and $\bar{S} \sim \varepsilon^2 T^2$, where $T$ is the period of the system response. Depending on the value of the $V$-number, the system response may feature periods of the external forcing, or multiples of forcing periods, or combinations of them. Accordingly, Fig. 1D shows a "-1.9" slope in the orbital frequency domain (though, again, all peaks in this domain are, in fact, created by the millennial forcing) and a "-2" slope for the millennial domain. We consider this result as a remarkable test in favor of the above hypothesis, i.e. $\bar{S} = \psi$ $(V, \varepsilon, T)$. Different aspects of glacial geometry such as area ($S$), ice thickness ($H \sim S^{1/4}$, Verbitsky, et al., 2018), glaciation horizontal span ($\sim S^{1/2}$), or ice volume ($HS \sim S^{5/4}$), may play a role in shaping climate conditions in a specific geographical point. Thus, corresponding power spectra of empirical records may have frequency slopes

ranging from "-5" ($f^{(-2)*5/4*2}$) to "-1" ($f^{(-2)*1/4*2}$). For multi-period forcing in a frequency domain $\Delta f$, the aggregate "hijacking" potential $\Delta S_A$ can be estimated as:

$$\Delta S_A = \frac{1}{\Delta f} \int \Delta S df = -\mu V \Delta f^{-1} \int \varepsilon(f)^2 f^{-2} df \qquad (2)$$

Accordingly, since amplitudes of high-frequency variability, $\varepsilon(f)$, may compensate for the frequency damping ($f^{-2}$), centennial, decennial, and perhaps even annual variations potentially may contaminate the spectrum throughout the millennial and multi-millennial range, and perturb ice age dynamics via two physical mechanisms : (a) centennial and millennial oscillations shift the mean state of the system, and (b) the sensitivity of ice sheets to the astronomical forcing depends on the system state.

**Discussion.** To our knowledge, only Le Treut and Ghil (1983) have previously adopted a deterministic framework to model a background spectrum connecting millennial to astronomical time scales. Unfortunately, their model did not generate credible ice age time series. The more common route for simulating the centennial and millennial spectrum is to introduce a stochastic forcing (e.g. Wunsch, 2003, Ditlevsen and Crucifix, 2017). Such stochastic forcing may in principle be justified by the existence of chaotic or turbulent motion in the atmosphere-ocean continuum. However, whether such forcing is large enough to integrate all the way up to time scales of several tens of thousands of years is speculative. The deterministic theory proposed here presents the advantage of using the non-linear character of ice sheet dynamics, which was derived naturally from the conservation laws and therefore provides a clear physical interpretation of the non-linear origin of the cascade. Our approach is thus remarkably parsimonious, because it requires no more physics than the minimum needed to explain ice ages, plus the existence of centennial or millennium modes of motions. The latter may very plausibly arise as modes of ocean motion (Dijkstra and Ghil, 2005, Peltier and Vettoretti, 2014). Of course, stochastic forcing may still be added, and its cumulated effects would then be estimated by equation (2).

In summary, using deterministic non-linear dynamical model of the global climate, we demonstrated that astronomical time-scale variability cannot be considered separately from millennial phenomena and that the ice dynamics is an effective vehicle for propagating high-frequency forcing into the orbital time scale. This may imply that the knowledge of millennial and centennial variability is needed to fully understand and replicate ice-age history. As we have seen, *increased* millennial variability *decreases* the length of the ice age cycles. However, the reverse is also true. This state of affairs generates a new hypothesis for the middle-Pleistocene transition: a decrease in millennial variability may have caused the lengthening of ice ages. The millennial variability can legitimately be modelled as a deterministic mode, which would allow us to come up with a specific explanation of how this variability may influence ice age dynamics. Hence our completely deterministic approach makes a physically justified alternative to a popular notion that the background spectrum is merely linearly integrated noise.

**Code and data availability.** The MatLab R2015b code and data to calculate model response to astronomical and millennial forcing (Verbitsky et al., 2019) are available at http://doi.org/10.5281/zenodo.2628310 (last access: 4 April 2019)

**Acknowledgements.** MC is funded by the Belgian National Fund of Scientific Research. Dmitry Volobuev is funded in part by the Russian Foundation for Basic Research, grant 19-02-00088-a. We are grateful to our reviewers Gerrit Lohmann and Niklas Boers for their helpful comments.

**Author contributions.** MV conceived the research and developed the model. MV and MC wrote the paper. DV developed the numerical scheme and MatLab code.

**Competing interests.** The authors declare that they have no conflict of interest.

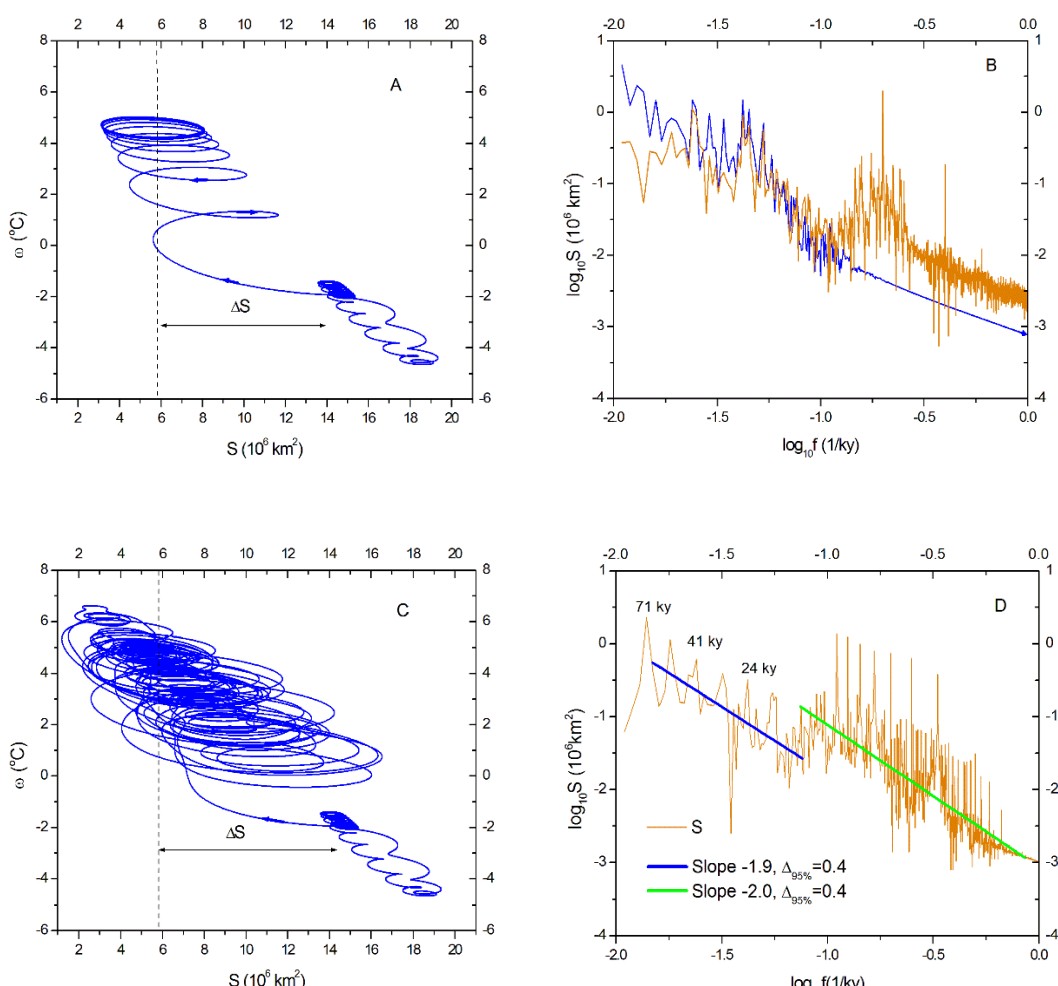

**Fig. 1 A**: The system response to pure 5-ky sinusoid of variable amplitude on a phase plane of glaciation area $S$ ($10^6$ km$^2$) *vs.* climate temperature $\omega$ ($^o$C); $\Delta S$-is disruption potential; **B**: Blue line represents reference system response to astronomical forcing (Verbitsky et al., 2018). Millennial forcing is absent here. Brown line shows the system response when the orbital forcing is combined with 5-ky sinusoid of the tenfold amplitude. The diagram is a linear amplitude spectrum on logarithmic scale; vertical axis measures

the amplitude of glacial area variations, $\log_{10} [\overline{S} (10^6$ km$^2)]$; horizontal axis is $\log_{10}[f (1/\text{ky})]$; **C**: Same as Section A but millennial forcing is formed by seven sinusoids of the same amplitude (~2.5 of the insolation forcing amplitude) and periods of 3-, 4-, 5-, 6-, 7-, 8-, and 9-ky. **D**: Same as Section B for multi- millennial forcing of Section C.

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
