# Peer review of "ESD Ideas: Propagation of high-frequency forcing to ice age dynamics"

_Earth System Dynamics, 2018_

## Referee Comment (RC1) · Lohmann (Referee) · 8 Dec 2018

**Review**

Verbitsky et al. (submitted to ESD) show evidence that short-term fluctuations can affect the long-term spectrum of climate variability of Pleistocene ice ages. In previous work (Saltzman and Verbitsky, 1994; Paillard and Parrenin, 2004; Verbitsky et al., 2018) the nature of Pleistocene ice ages was explored using simplified/conceptual climate models. Some papers call for an ice-carbon dioxide oscillator to produce 1000 kyr cycles, the recent Verbitsky et al. (2018) work does not require a nonlinear response of the carbon cycle. The underlying dynamical system has 11 parameters, linked to empirical data of present ice sheets and others are linked to ice sheet and climate models.

[Figure]

One beauty in the approach is that the system behavior can be described by the dimensionless V number (Verbitsky et al., 2018). Using the $\Pi$-theorem (Buckingham, 1914), the authors derive a scaling for the disruption potential $\Delta S$ for the phase space analysis (eq. 1). The authors find that when adding deterministic millennial modes to the system, the spectrum can be dramatically different.

**Recomendation:**

The manuscript provides a new perspective of Pleistocene ice ages by using a conceptual model of Verbitsky et al. (2018). It shows a potential coupling between orbital and centennial-millennial variability. The paper needs only moderate revisions before acceptance. In the following, I list some specific comments.

**Specific comments:**

- The wording "non-linear system response". Due to the Northern Hemisphere summer forcing (Berger, 1978), the system receives already a strong non-linearity. Please clarify this somewhere.

- Reference to the $\Pi$-theorem (Buckingham, 1914), the idea goes even further back with Bertrand (1878). Though the Barenblatt (2003) book is a potential reference, I suggest using the older literature here.

- The sentence "This observation makes centennial, decennial, and maybe even annual variations potentially able to contaminate the spectrum throughout the millennial and multi-millennial range and perturb ice age dynamics." (lines 36 ff.) is essential for the conclusions. I would ask to substantiate it more with physics.

- The authors proposed that their deterministic approach has advantages to show that the forcing propagates upscale. I find the deterministic forcing a little arbitrary. I cannot follow the sentence " ... presents the advantage of using the non-linear character of ice sheet dynamics, which was derived naturally from the conservation laws ...". Given the stochastic nature of the millennial variability (Ditlevsen, 1999), the paper would benefit from an additional stochastic analysis which could be added. You mention that "the dynamics at the centennial, millennial, and astronomical time scales should not be considered separately. "

- Related to the last point: The single 5-ky high-amplitude sinusoid, moving the system into the phase-plane domain of higher temperatures and lower ice volume, is not motivated. Known modes are in the ∼2.5, 0.9, and 0.5 ky-bands (e.g., Dima and Lohmann, 2008). The periods spread between 3 ky and 9 ky are not really motivated. You may also mention that the mechanism you found is probably different from "noise-induced transitions" where the stochastic forcing generates new equilibria, which do not have a deterministic counterpart.

- The implications are only roughly sketched. Would your result imply that we need high-resolution data to understand the variations on orbital time scales? This is, of course, difficult because of the limited space and references allowed here.

- Please check the internal consistency of notations, e.g. ky and ka.

**References**

Berger, A. L.: Long-term variations of daily insolation and Quaternary climatic changes, J. Atmos. Sci., 35, 2362-2367, 1978.

Bertrand J., 1878: Sur l'homogénété dans les formules de physique. Comptes rendus 86 (15), 916-920.

Buckingham E., 1914: On physically similar systems: illustrations of the use of dimensional equations. Physical Review 4, Nr. 4, 1914, S. 345–376.

Dima, M., and Lohmann, G., 2008: Conceptual model for millennial climate variability: a possible combined solar-thermohaline circulation origin for the ∼1,500-year cycle. Climate Dynamics. 32 (2-3), 301-311.

Ditlevsen, P.D., 1999: Observation of a-stable noise induced millennial climate changes from an ice-core record. GRL 26 (10), 1441-1444.

Saltzman, B. and Verbitsky, M. Y.: Multiple instabilities and modes of glacial rhythmicity in the PlioPleistocene: a general theory of late Cenozoic climatic change, Clim. Dynam., 9, 1-15, 1993.

Paillard, D., 2015: Quaternary glaciations: from observations to theories, Quat. Sci. Rev. 107, 11-24.

Paillard, D. and Parrenin, F., 2004: The Antarctic ice sheet and the triggering of deglaciations. Earth Plan. Sci. Lett. 227, 263-271.

Timmermann, A., and Lohmann, G., 2000: Noise-Induced Transitions in a simplified model of the thermohaline circulation, J. Phys. Oceanogr. 30 (8), 1891-1900.

Verbitsky, M. Y., Crucifix, M., and Volobuev, D. M., 2018: A theory of Pleistocene glacial rhythmicity, Earth Syst. Dynam., 9, 1025-1043.

Gerrit Lohmann

---

## Referee Comment (RC2) · Niklas Boers (Referee) · 13 Feb 2019

Summary: In a sense, this manuscript is an extension or sequel to an ESD paper by the same authors from 2018. The main conclusion, derived from additional simulations with the model introduced in that paper, is that millennial and / or centennial scale processes can have a strong effect on ice age dynamics. It is submitted as an ESD ideas paper, and is consequently quite brief on some aspects. At several points, I would actually say that more detailed explanations would be helpful, depending on the space requirements for ideas, maybe even turning this into a 'normal' submission? Please see the following

Questions and Comments:

[Figure]

p1 l43: mid-June insolation at which latitude?

p2 l5: I think it is not explained why, in your model, the strong millennial-scale forcing leads to this specific change in the fixed point (in particular, warmer and less ice). Could you elaborate on the underlying mechanism? l11ff: I'm not sure if I understand what you write under part D: First, do you mean that the original frequencies of low-amplitude sinosoids are preserved by the model? Second, I thought that this case would refer to only periodic (and no astronomical) forcing, how can precession and obliquity be overlapped? See also the corresponding figure panel D.

l18ff: Could you elaborate how you use the Buckingham theorem to obtain this specific scaling relation? In particular, why does the amplitude have units km/kyr? You say previously that the amplitude of the periodic forcing is of similar amplitude as the insolation. Also, it would be good to carry out at a level of detail that allows everyone to understand why the exponents are fixed to -2, because this is crucial later on. Ideally, there would be a plot showing (from simulation data) that \Delta S is a quadratic function of (epsilon T).

l29: I'm not sure the 'brown' (red?) amplitude spectrum really has slope -2. Have you tried to make a linear fit for comparison?

l30: I don't understand where the different exponents come from; in particular, how does the "-5 to -1" range exactly relate to to exponents given in the lines above?

l34: I would suggest to make the frequency dependency of epsilon explicit

l35: Can you explain your interpretation of Eq (2), please? In principle, high frequencies are damped by the f^-2 term. Your main conclusion, that ice age dynamics can be affected by centennial time scales (in your model), is evident from Fig.1A, but I find it hard to infer this from Eq. (2) alone. It clear that there is interaction between the slow and the fast scales, but it's not clear how strong, because here it really depends on epsilon.

[Figure]

Figure: - Would it be possible to provide them in higher resolution? - I would suggest to interchange panels B and C - remove the word "Section" from the caption - i think it would be better to use the same axes for panels A and C

Technical corrections:

p1 l33: ... state of rest of the climate,...? l37: ... of the positive climate feedback to the negative glaciation feedback l40: ... to an - even modest - increase ... l42: ..., following standard practise, ...? l44: ...with sinosoids of 5-ky periodicity and variable amplitude. -> Could you be more specific regarding the ' variable' amplitude? l45: ... 5yr-periodic sinosoids of amplitude about ... p2 l17: ... given a periodic forcing ... l21: according to the $\pi$-theorem l44 ... forcing is large enough ...

Very best,

Niklas

---

## Author Comment (AC2) · 28 Feb 2019

**Interactive comment* on "ESD Ideas: Propagation of high-frequency forcing to ice age dynamics" *by* Mikhail Verbitsky et al.**

**M. Verbitsky**

verbitskys@gmail.com

**Niklas Boers,**

Dear Dr. Boers,

Thank you for your detailed review and constructive suggestions. The following is our response to your Questions and Comments. We believe that all requested clarifications can be done within ESD Ideas format limit.

**Question or Comment:** p1 l43: mid-June insolation at which latitude?
**Answer:** at 65$^o$N

**Action:** This will be clarified

**Question or Comment:** p2 l5: I think it is not explained why, in your model, the strong millennial-scale forcing leads to this specific change in the fixed point (in particular, warmer and less ice). Could you elaborate on the underlying mechanism? l11ff: I'm not sure if I understand what you write under part D: First, do you mean that the original frequencies of low amplitude sinusoids are preserved by the model? Second, I thought that this case would refer to only periodic (and no astronomical) forcing, how can precession and obliquity be overlapped? See also the corresponding figure panel D.
**Answer:** Because of the system's non-linearity, the response trajectory to negative forcing is not symmetric to the trajectory generated by a positive forcing. This leads to a shift of the time-mean ice-sheet area and temperature (a "hijacking" effect). When the system is "hijacked" by several sinusoids, millennial forcing is capable of making combined periods that are close to the orbital periods, e.g., periods of precession and obliquity. Specifically, millennial frequencies form a beating modulated by a low-frequency envelope. The model then has the capacity to demodulate a beating signal and respond to its modulating envelope. For example, 41-ky mode in Fig. 1D, which one might be tempted to attribute to obliquity, is in fact the demodulation of the envelope generated by the interplay of the 6-ky and 7-ky forcing sinusoids: 1/41≈1/6-1/7. Thus two millennial frequencies have created a low frequency forcing with a period similar to the orbital period, i.e. obliquity. The idea of combined periods is not new, this notion of 'combination of harmonics' was emphasized by Le Treut and Ghil (1983), who suggested that

the 100-ky period characterizing the late Pleistocene glaciation came from the precession beating (1/100 ≈ 1/19 - 1/23). We demonstrated here (and this is, indeed, new finding) that the periods of the orbital domain (or better to say, of the domain traditionally "reserved" for orbital periods) can be produced by the millennial forcing. On Figure 1D, you can see both original millennial periods and demodulated-envelope periods.

**Action:**  We will add this discussion into the text.

**Question or Comment:** Could you elaborate how you use the Buckingham theorem to obtain this specific scaling relation? In particular, why does the amplitude have units km/kyr? You say previously that the amplitude of the periodic forcing is of similar amplitude as the insolation. Also, it would be good to carry out at a level of detail that allows everyone to understand why the exponents are fixed to -2, because this is crucial later on. Ideally, there would be a plot showing (from simulation data) that \Delta S is a quadratic function of (epsilon T).
**Answer:** We describe here our reasoning as it is applied to the variable S. Similar rationale can be also applied to the disruptive potential $\Delta S$.

The statement S = ɸ (V, ε, T) is not, indeed, an exact solution of the system of differential equations but a hypothesis that has been inspired by a significant number of numerical experiments we conducted with our model ("numerical observations", so to say). It provides a starting point for reasoning and also allows estimating the order of magnitude of scaling relationships in the model. Indeed, the V-number is a dimensionless combination of 8 model parameters. On the other hand, the external (astronomical or millennial) forcing of amplitude ε is introduced in our model as a component of the ice sheet surface mass balance and, therefore, it  has the same dimension as ice accumulation/ablation rate: km/ky (Verbitsky et al, 2018; equations 18, 19). T is the forcing period, in ky. If the statement S = ɸ (V, ε, T) is true then, according to π-theorem, $S/(\varepsilon^2 T^2)$ = F (V).  While the F(V) function needs to be determined experimentally, it definitely doesn't depend on T. Hence, the frequency slope of the amplitude spectrum of the system response (in terms of S-variable) should be close enough to "-2".

In summary, the hypothesis S = ɸ (V, ε, T) needs indeed to be tested. Your next question (and our corresponding answer) concerns this test.

**Action:**  (a) We will clarify the dimension of the amplitude of the astronomical and millennial forcing, and (b ) we will conduct additional experiments to illustrate that $\Delta S$ is indeed proportional to $(\varepsilon T)^2$.

**Question or Comment:** l29: I'm not sure the 'brown' (red?) amplitude spectrum really has slope -2. Have you tried to make a linear fit for comparison?

**Answer:** In the modified Figure 1D below, we present amplitude spectrum of the system response to millennial forcing made of seven sinusoids of the same amplitude and periods of 3-, 4-, 5-, 6-, 7-, 8-, and 9-ky. The linear fit shows a "-1.8" slope in the orbital frequency domain (though, again, all peaks in this domain are, in fact, created by the millennial forcing) and a "-2" slope for the millennial domain. We consider this result as a remarkable test in favor of the above hypothesis, i.e. S = ɸ (V, ε, T).

[Figure]

**Action:** We will update Fig. 1D to include linear fit as it is shown above.

**Question or Comment:** l30: I don't understand where the different exponents come from; in particular, how does the "-5 to -1" range exactly relate to exponents given in the lines above?

**Answer:** Ice thickness H is proportional to the glaciation area as $S^{(1/4)}$ (Verbitsky et al, 2018; equation 5). Accordingly, ice volume, HS, is proportional to $S^{(5/4)}$. If a component of the climate system depends on the glaciation area as $S^{\alpha}$ and S is proportional to $f^{(-2)}$, then the amplitude spectrum of this variable will be proportional to $f^{(-2\alpha)}$, and the power spectrum will be proportional to $f^{(-4\alpha)}$. For $\alpha=5/4$ (responding to volume) it gives a frequency slope of "-5" and for $\alpha=1/4$ (responding to height) it gives a frequency slope of "-1".

**Action:** We will clarify this in the text

**Question or Comment:** l34: I would suggest to make the frequency dependency of epsilon explicit

**Answer:** We agree that it would be helpful.

**Action:** Equation (2) will be modified accordingly

**Question or Comment:** l35: Can you explain your interpretation of Eq (2), please? In principle, high frequencies are damped by the f^-2 term. Your main conclusion, that ice age dynamics can be affected by centennial time scales (in your model), is evident from Fig.1A, but I find it hard to infer this from Eq. (2) alone. It (is) clear that there is interaction between the slow and the fast scales, but it's not clear how strong, because here it really depends on epsilon.

**Answer:** Two physical mechanisms can be of particular importance for propagating high frequencies upscale: (a) centennial and millennial oscillations shift the mean state of the system, and (b) the sensitivity of the ice sheets to the astronomical forcing depends on the system state. Your observation that amplitudes of high-frequency variability may compensate for frequency damping ($f^{-2}$) is correct.

**Action:** We will provide additional discussion in the text

**Question or Comment:** Figure: - Would it be possible to provide them in higher resolution? - I would suggest to interchange panels B and C - remove the word "Section" from the caption - i think it would be better to use the same axes for panels A and C

**Answer:** All of the above can be done.

**Action:** We will provide higher quality pictures.

**Question or Comment:** Technical corrections: p1 l33: ... state of rest of the climate,...? l37: ... of the positive climate feedback to the negative glaciation feedback l40: ... to an - even modest - increase ... l42: ..., following
standard practise, ...? l44: ...with sinosoids of 5-ky periodicity and variable amplitude.-> Could you be more specific regarding the ' variable' amplitude? l45: ... 5yr-periodic sinosoids of amplitude about ... p2 l17: ... given a periodic forcing ... l21: according to the π-theorem l44 ... forcing is large enough ...

**Answer:** Thank you for noticing

**Action:** Everything will be fixed

**References**

Le Treut, H. and Ghil, M.: Orbital forcing, climatic interactions, and glaciation cycles. Journal of Geophysical Research: Oceans, 88(C9), 5167-5190, 1983

Verbitsky, M. Y., Crucifix, M., and Volobuev, D. M.: A theory of Pleistocene glacial rhythmicity, Earth Syst. Dynam., 9, 1025-1043, https://doi.org/10.5194/esd-9-1025-2018, 2018

---

## Author Response (AR1)

**Interactive comment* on "ESD Ideas: Propagation of high-frequency forcing to ice age dynamics" by Mikhail Verbitsky et al.**

**M. Verbitsky**

**verbitskys@gmail.com**

Gerrit Lohmann, Received and published: 8 December 2018

Dear Professor Lohmann,

Thank you for your insightful review and practical suggestions. The following is our response to your specific comments:

**Comment:** "The wording "non-linear system response". Due to the Northern Hemisphere summer forcing (Berger, 1978), the system receives already a strong nonlinearity. Please clarify this somewhere"

**Answer:** We concede that the phrasing was unfortunate. When we say "This non-linear system response has a dramatic effect on ice-age dynamics", we mean that the system responds to a millennial-period sinusoid by a shift of the time-mean ice-sheet area and temperature, and that this shift depends non-linearly on the amplitude of the forcing.

**Action: We will articulate this thinking more clearly - p.2 line 7**

**Comment:** "Reference to the Π-theorem (Buckingham, 1914), the idea goes even further back with Bertrand (1878). Though the Barenblatt (2003) book is a potential reference, I suggest using the older literature here" **Answer:** We agree.

**Action: We will reference $\pi$ -theorem to Buckingham (1914) – p. 2 line 30**

**Comment:** "The sentence "This observation makes centennial, decennial, and maybe even annual variations potentially able to contaminate the spectrum throughout the millennial and multi-millennial range and perturb ice age dynamics." (lines 36 ff.) is essential for the conclusions. I would ask to substantiate it more with physics." **Answer:** We agree that this observation deserves more discussion.

Action: We will articulate our conclusion with more details. Specifically we will emphasize that (a) centennial and millennial oscillations shift the mean state of the system, and (b) the sensitivity of ice sheets to the astronomical forcing depends on the

system state. Taken together, these two observations show how centennial and millennial variability can perturb ice-age dynamics and, hence, contaminate the spectrum of variability. – **p. 3 lines 6-8**

**Comment:** "The authors proposed that their deterministic approach has advantages to show that the forcing propagates upscale. I find the deterministic forcing a little arbitrary. I cannot follow the sentence "... presents the advantage of using the non-linear character of ice sheet dynamics, which was derived naturally from the conservation laws ...". Given the stochastic nature of the millennial variability (Ditlevsen, 1999), the paper would benefit from an additional stochastic analysis which could be added. You mention that "the dynamics at the centennial, millennial, and astronomical time scales should not be considered separately"

**Answer:** It is correct that several authors (including one of us) have adopted noise models to express the effects of chaotic fluctuations --- with, generally, a reference to Hasselman's theory. Stochastic forcing do indeed provide a potentially fruitful approach to explain the background spectrum, with the reservation that we still have no good theory to determine how to quantify this forcing. A deterministic forcing provides other benefits which we wanted to take advantage of here. First, it allows us clearly to identify the non-linear origin of the cascade ---- while a stochastic forcing may simply be integrated linearly. Second, millennial variability can legitimately be modelled as a deterministic mode, which allows us to come up with a specific explanation of how this variability may influence ice age dynamics.

**Action: We will add more discussion about deterministic versus stochastic approaches – p. 3 lines 17-18, 30-33**

**Comment:** "Related to the last point: The single 5-ky high-amplitude sinusoid, moving the system into the phase-plane domain of higher temperatures and lower ice volume, is not motivated. Known modes are in the 2.5, 0.9, and 0.5 ky-bands (e.g., Dima and Lohmann, 2008). The periods spread between 3 ky and 9 ky are not really motivated. You may also mention that the mechanism you found is probably different from "noise-induced transitions" where the stochastic forcing generates new equilibria, which do not have a deterministic counterpart"

**Answer: T**he 5-ky sinusoid and multiple sinusoids of periods spread between 3-ky and 9ky are, indeed, arbitrary. We used them to demonstrate the "hijacking" phenomenon as well as ability of high frequencies to form a beating modulated by a low-frequency envelope and the model's capacity to demodulate a beating signal and to respond to its modulating envelope. This said, perhaps we should also acknowledge that identifying precise mode frequencies from time series analysis is not straightforward either. Time series analysis often relies on assumptions of stationarity and Gaussianity which are not always well verified. This motivates our original choice of using generic frequencies to represent the phenomenon of millennial variability. Hence, we understand the reviewer's concern. To address this concern, we used scaling arguments to anticipate the disruptive effect of forcing at other periods.

**Action: We will complement our research with additional experiments with known modes of millennial variability and describe the results in the text – **p. 2 lines 35-40**

**Comment:** "The implications are only roughly sketched. Would your result imply that we need high-resolution data to understand the variations on orbital time scales? This is, of course, difficult because of the limited space and references allowed here" **Answer:** One implication of our study is to formulate a realistic, physically justified alternative to the notion that the background spectrum is merely linearly-integrated noise. In doing so, we are alerting the reader to the potential pitfalls of the classical time-scale separation hypothesis, used to justify delivering separate explanations for DO-oscillations and ice ages.

**Action: We will add to implications discussions - p.3 lines 18-33**

**Comment:** "Please check the internal consistency of notations, e.g. ky and ka" **Answer:** Thank you for noticing.

**Action: Done**

Thank you for your detailed review and constructive suggestions. The following is our response to your Questions and Comments. We believe that all requested clarifications can be done within ESD Ideas format limit.

**Question or Comment:** p1 I43: mid-June insolation at which latitude? **Answer:** at 65°N

**Action: This will be clarified - p.1 lines 46, 50**

Question or Comment: p2 I5: I think it is not explained why, in your model, the strong millennial-scale forcing leads to this specific change in the fixed point (in particular, warmer and less ice). Could you elaborate on the underlying mechanism? I11ff: I'm not sure if I understand what you write under part D: First, do you mean that the original frequencies of low amplitude sinusoids are preserved by the model? Second, I thought that this case would refer to only periodic (and no astronomical) forcing, how can precession and obliquity be overlapped? See also the corresponding figure panel D. **Answer:** Because of the system's non-linearity, the response trajectory to negative forcing is not symmetric to the trajectory generated by a positive forcing. This leads to a shift of the time-mean ice-sheet area and temperature (a "hijacking" effect). When the system is "hijacked" by several sinusoids, millennial forcing is capable of making combined periods that are close to the orbital periods, e.g., periods of precession and obliguity. Specifically, millennial frequencies form a beating modulated by a lowfrequency envelope. The model then has the capacity to demodulate a beating signal and respond to its modulating envelope. For example, 41-ky mode in Fig. 1D, which one might be tempted to attribute to obliquity, is in fact the demodulation of the envelope generated by the interplay of the 6-ky and 7-ky forcing sinusoids: 1/41≈1/6-1/7. Thus two millennial frequencies have created a low frequency forcing with a period similar to the orbital period, i.e. obliguity. The idea of combined periods is not new, this notion of

'combination of harmonics' was emphasized by Le Treut and Ghil (1983), who suggested that the 100-ky period characterizing the late Pleistocene glaciation came from the precession beating ( $1/100 \approx 1/19 - 1/23$ ). We demonstrated here (and this is, indeed, new finding) that the periods of the orbital domain (or better to say, of the domain traditionally "reserved" for orbital periods) can be produced by the millennial forcing. On Figure 1D, you can see both original millennial periods and demodulated-envelope periods.

**Action: We will add this discussion into the text – p.2 lines 17-22**

**Question or Comment:** Could you elaborate how you use the Buckingham theorem to obtain this specific scaling relation? In particular, why does the amplitude have units km/kyr? You say previously that the amplitude of the periodic forcing is of similar amplitude as the insolation. Also, it would be good to carry out at a level of detail that allows everyone to understand why the exponents are fixed to -2, because this is crucial later on. Ideally, there would be a plot showing (from simulation data) that \Delta S is a quadratic function of (epsilon T).

**Answer:** We describe here our reasoning as it is applied to the variable S. Similar rationale can be also applied to the disruptive potential  $\Delta$ S.

The statement  $S = \phi$  (V,  $\varepsilon$ , T) is not, indeed, an exact solution of the system of differential equations but a hypothesis that has been inspired by a significant number of numerical experiments we conducted with our model ("numerical observations", so to say). It provides a starting point for reasoning and also allows estimating the order of magnitude of scaling relationships in the model. Indeed, the V-number is a dimensionless combination of 8 model parameters. On the other hand, the external (astronomical or millennial) forcing of amplitude  $\varepsilon$  is introduced in our model as a component of the ice sheet surface mass balance and, therefore, it has the same dimension as ice accumulation/ablation rate: km/ky (Verbitsky et al, 2018; equations 18, 19). T is the forcing period, in ky. If the statement  $S = \phi$  (V,  $\varepsilon$ , T) is true then, according to  $\pi$ -theorem,  $S/(\varepsilon^2T^2) = F$  (V). While the F(V) function needs to be determined experimentally, it definitely doesn't depend on T. Hence, the frequency slope of the amplitude spectrum of the system response (in terms of S-variable) should be close enough to "-2".

In summary, the hypothesis S =  $\phi$  (V,  $\varepsilon$ , T) needs indeed to be tested. Your next question (and our corresponding answer) concerns this test.

Action: (a) We will clarify the dimension of the amplitude of the astronomical and millennial forcing, and (b) we will conduct additional experiments to illustrate that  $\Delta S$  is indeed proportional to  $(\epsilon T)^2 - p.2$  lines 28-29, 35-40

**Question or Comment:** I29: I'm not sure the 'brown' (red?) amplitude spectrum really has slope -2. Have you tried to make a linear fit for comparison?

**Answer:** In the modified Figure 1D below, we present amplitude spectrum of the system response to millennial forcing made of seven sinusoids of the same amplitude and periods of 3-, 4-, 5-, 6-, 7-, 8-, and 9-ky. The linear fit shows a "-1.8" slope in the orbital frequency domain (though, again, all peaks in this domain are, in fact, created by the millennial forcing) and a "-2" slope for the millennial domain. We consider this result as a remarkable test in favor of the above hypothesis, i.e.  $S = \phi$  (V,  $\varepsilon$ , T).

**Action: See new Fig. 1D**

**Question or Comment:** I30: I don't understand where the different exponents come from; in particular, how does the "-5 to -1" range exactly relate to exponents given in the lines above?

**Answer:** Ice thickness H is proportional to the glaciation area as  $S^{(1/4)}$  (Verbitsky et al, 2018; equation 5). Accordingly, ice volume, HS, is proportional to  $S^{(5/4)}$ . If a component of the climate system depends on the glaciation area as  $S^{\alpha}$  and S is proportional to  $f^{(-2)}$ , then the amplitude spectrum of this variable will be proportional to  $f^{(-2\alpha)}$ , and the power spectrum will be proportional to  $f^{(-4\alpha)}$ . For  $\alpha=5/4$  (responding to volume) it gives a frequency slope of "-5" and for  $\alpha=1/4$  (responding to height) it gives a frequency slope of "-1".

**Action: We will clarify this in the text – p.3 line 1**

**Question or Comment:** I34: I would suggest to make the frequency dependency of epsilon explicit

Answer: We agree that it would be helpful.

**Action: See new Equation (2)**

**Question or Comment:** 135: Can you explain your interpretation of Eq (2), please? In principle, high frequencies are damped by the f-2 term. Your main conclusion, that ice age dynamics can be affected by centennial time scales (in your model), is evident from Fig.1A, but I find it hard to infer this from Eq. (2) alone. It (is) clear that there is interaction between the slow and the fast scales, but it's not clear how strong, because here it really depends on epsilon.

**Answer:** Two physical mechanisms can be of particular importance for propagating high frequencies upscale: (a) centennial and millennial oscillations shift the mean state of the system, and (b) the sensitivity of the ice sheets to the astronomical forcing depends on the system state. Your observation that amplitudes of high-frequency variability may compensate for frequency damping ( $f^{-2}$ ) is correct.

**Action: We will provide additional discussion in the text - p 3 lines 3-8**

**Question or Comment:** Figure: - Would it be possible to provide them in higher resolution? - I would suggest to interchange panels B and C - remove the word "Section" from the caption - i think it would be better to use the same axes for panels A and C **Answer:** All of the above can be done.

**Action: Done, see new Figure 1.**

**Question or Comment:** Technical corrections: p1 I33: ... state of rest of the climate,...? I37: ... of the positive climate feedback to the negative glaciation feedback I40: ... to an even modest - increase ... I42: ..., following standard practise, ...? I44: ...with sinosoids of 5-ky periodicity and variable amplitude.-> Could you be more specific regarding the ' variable' amplitude? I45: ... 5yr-periodic sinosoids of amplitude about ... p2 I17: ... given a periodic forcing ... I21: according to the  $\pi$ -theorem I44 ... forcing is large enough ... **Answer:** Thank you for noticing

**Action: Done**

**References**

[revised manuscript text omitted]